# Sulfenate anions as organocatalysts for benzylic chloromethyl coupling polymerization via C=C bond formation

Minyan Li[1], Simon Berritt [1], Carol Wang[1], Xiaodong Yang [1,2], Yang Liu[1], Sheng-Chun Sha[1], Bo Wang[1], Rui Wang[3], Xuyu Gao[4], Zhanyong Li[5], Xinyuan Fan[3], Youtian Tao[4] & Patrick J. Walsh [1,3]

Organocatalytic polymerization reactions have a number of advantages over their metal-catalyzed counterparts, including environmental friendliness, ease of catalyst synthesis and storage, and alternative reaction pathways. Here we introduce an organocatalytic polymerization method called benzylic chloromethyl-coupling polymerization (BCCP). BCCP is catalyzed by organocatalysts not previously employed in polymerization processes (sulfenate anions), which are generated from bench-stable sulfoxide precatalysts. The sulfenate anion promotes an umpolung polycondensation via step-growth propagation cycles involving sulfoxide intermediates. BCCP represents an example of an organocatalyst that links monomers by C=C double bond formation and offers transition metal-free access to a wide variety of polymers that cannot be synthesized by traditional precursor routes.

[1] Roy and Diana Vagelos Laboratories, Penn/Merck Laboratory for High-Throughput Experimentation, Department of Chemistry, University of Pennsylvania, 231 South 34th Street, Philadelphia, Pennsylvania 19104, USA. [2] Key Laboratory of Medicinal Chemistry for Natural Resources, Ministry of Education and Yunnan Province, School of Chemical Science and Technology, Yunnan University, Kunming 650091, P. R. China. [3] Institute of Advanced Synthesis, School of Chemistry and Molecular Engineering, Jiangsu National Synergetic Innovation Center for Advanced Materials, Nanjing Tech University, 30 South Puzhu Road, Nanjing 211816, China. [4] Key Laboratory of Flexible Electronics (KLOFE) and Institute of Advanced Materials (IAM), Nanjing Tech University, 30 South Puzhu Road, Nanjing 211816, China. [5] Key Laboratory of Functional Polymer Materials, Ministry of Education, Institute of Polymer Chemistry, Nankai University, Tianjin 300071, China. Correspondence and requests for materials should be addressed to P.J.W. (email: pwalsh@sas.upenn.edu)

I nnovations in polymer chemistry and materials science often have their genesis in the introduction of small molecule catalysts[1,2]. This is particularly true in the developing field of organocatalytic polymerization chemistry[3,4]. Organocatalytic polymerization reactions have a number of advantages over their metal-catalyzed counterparts, including environmental friendliness, reduced toxicity and cost, ease of catalyst synthesis and storage, and access to alternative reaction pathways. Furthermore, organocatalysts circumvent problems caused by metal residue contamination of polymers, which can severely limit biomedical and electronic applications, and complicate polymer purification and processing[3,4]. The majority of organocatalytic polymerizations involve ring-opening polymerizations using cyclic esters, carbonates, ethers, siloxanes, anhydrides, and phosphoesters[4].

Herein we introduce a class of organocatalytic polymerization processes termed benzylic chloromethyl-coupling polymerization (BCCP). BCCP represents the application of sulfenate anion organocatalysts to polymerization processes. The sulfenate anion-catalyzed process proceeds via an umpolung mechanism and represents a rare example of an organocatalysts that enchains monomers by C=C bond formation[5,6]. Design of BCCP is validated in the context of poly(m-phenylene vinylene) (P$^m$PV) synthesis. In this study, P$^m$PV's with $M_n$ as high as 17,400 Da and with very high trans-selectivity are obtained. To demonstrate the mechanistic distinctness of BCCP, a non-conjugated polymer bearing quaternary –C(CF$_3$)$_2$ spacers between stilbene units in the polymer backbone is synthesized. Moreover, two alternating co-polymers as representatives of poly[(1,3-phenylene vinylene)-alt-arene]s and poly[(1,3-phenylene vinylene)-alt-(1,4-phenylene vinylene)] (P$^m$PV$^p$PV)s are synthesized. Nuclear magnetic resonance (NMR) spectra, thermal, photophysical, electrochemical, and charge transport properties of the above mentioned co-polymers are characterized. The polymers reported herein cannot be prepared by classic precursor routes (Gilch, Wesling, and Vanderzande methods).

## Results

**Design of the BCCP.** Sulfenate anions (ArSO$^−$) are highly reactive intermediates in biological chemistry and in organic reactions[7–10]. We recently disclosed that sulfenate anions can act as organocatalysts and reported their ability to catalytically dehydrocouple benzyl halides under basic conditions to yield trans-stilbenes (Fig. 1a)[11] and their application to catalytic cross-coupling of benzyl chlorides with benzaldehyde derivatives to produce diarylacetylenes[12]. The efficiency of sulfenate anion catalysts in these reactions, and their high selectivity for formation of trans-stilbenes, inspired us to explore their potential

**Fig. 1** Sulfenate anion-catalyzed reactions. **a** Dehydrocoupling of benzyl chlorides to produce stilbenes. **b** Generic representation of the benzylic chloromethyl-coupling polymerization (BCCP), where the blue box represents an aromatic system or tethered aromatic rings. **c** Fundamental steps in the organocatalytic BCCP

in polymerization reactions. We hypothesized that substrates bearing two benzylic chloromethyl groups would be suitable monomers for polymerization. The benzylic chloromethyl substituents could be located on the same aromatic system or on different aromatic rings separated by linking groups, as represented in Fig. 1b.

Based on this hypothesis, we designed the 1,3-bis(chloromethyl)benzene monomer **A** (Fig. 1c). We envisioned that the sulfenate anion would react with monomer **A** via an $S_N2$ reaction to generate sulfoxide **B**. In the presence of base, sulfoxide **B** is reversibly deprotonated to generate carbanion **C**. Anion **C** is a reactive nucleophile and undergoes $S_N2$ with monomer **A** to form the first C–C bond. Base promoted E2 elimination of intermediate **D** provides dimer **E** and liberates the sulfenate anion to further catalyze the polycondensation of **E**. Notably, the product is a $P^mPV$, which is an important class of organic semiconductors with applications in optoelectronics, such as organic light-emitting diodes (OLEDs), solar cells, organic lasers, sensors, and displays[13–18]. Although the synthesis of PPV's has been developed, including precursor routes[19–23], olefin metathesis polymerizations[23], nucleophilic condensations[24–29], and cross-coupling polymerizations[30–36], to the best of our knowledge this is a unique organocatalytic method for the synthesis of this important class of polymers. Moreover, $P^mPV$ is a challenging target, because *meta*-linkages preclude formation of

quinodimethane intermediates, prohibiting classic PPV precursor routes (Gilch, Wesling and Vanderzande methods)[33,37–40].

**Optimization of BCCP with monomer M1.** Starting from the optimized coupling of benzyl chlorides used in our stilbene synthesis (Fig. 1a)[11,] we selected cyclopentyl methyl ether (CPME) as solvent and KOtBu as base at 80 °C, to optimize the polymerization of monomer **M1** (Fig. 2). A long alkyl chain was introduced onto the $P^mPV$ backbone to assure the resulting polymer **P1** has good solubility in common organic solvents. Initial reactions were conducted in 24-well plates on 10 μmol scale by adapting small molecule high-throughput experimentation (HTE)[41–48] techniques to polymerizations (see Supplementary Method, High-Throughput Experimentation screenings for polymerization for full details). As shown in Fig. 2a, we initially focused on air-stable benzylic sulfoxide catalysts (**1–9**) ArSOCH₂Ph with various Ar–S groups and one precatalyst (**10**) with 4 catalyst loadings (10, 7.5, 5.0, and 2.5 mol %). Reactions were heated for 24 h at 80 °C followed by cooling and work up by addition of 10 μL of water and removal of the volatile materials. Next, CHCl₃ was added to each well to dissolve the products followed by cold methanol to precipitate the solid polymer. Finally, filtration of the solid, dissolution in tetrahydrofuran (THF) and analysis by gel permeation chromatography (GPC)

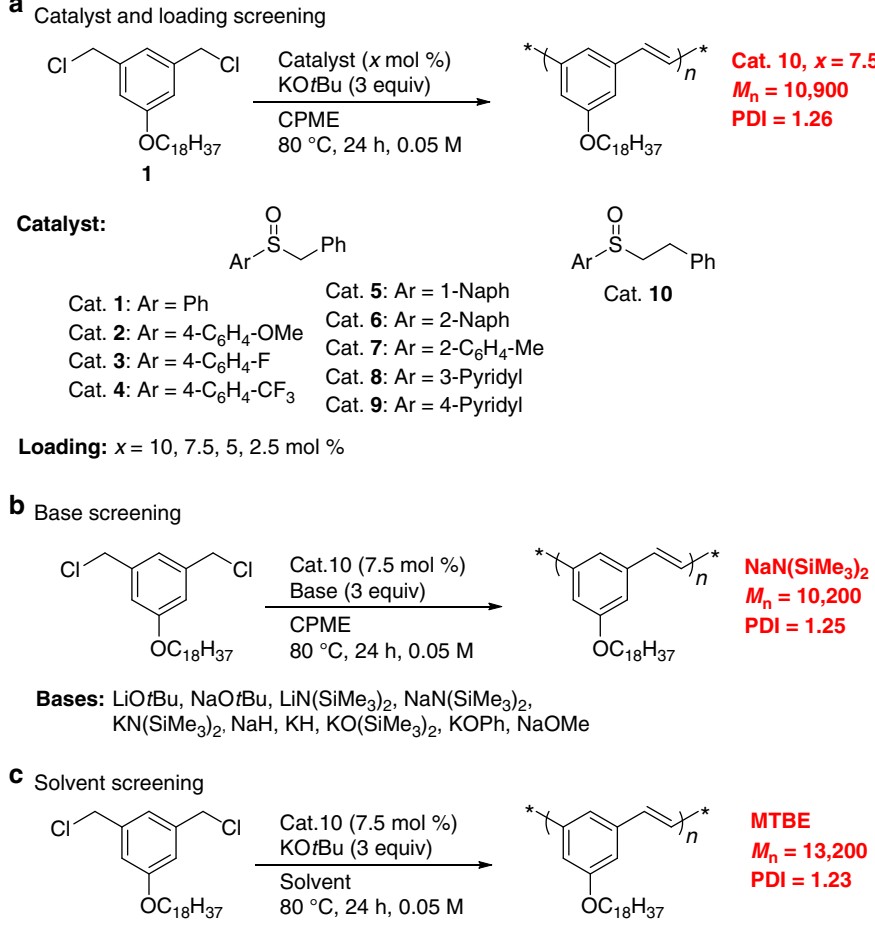

**Fig. 2** Optimization by HTE Screening. **a** Precatalysts employed leading to sulfoxide **10** for further studies. **b** Additional base screen indicated that NaN(SiMe₃)₂ ($M_n$ = 10,200) gave the best results, but lower than KOtBu ($M_n$ = 10,900) of panel **a**. **c** Additional solvent screen, wherein MTBE gave the best results ($M_n$ = 13,200) higher than CPME of panel **a**. See Supplementary Table 3 for all results

**Table 1 BCCP optimization**

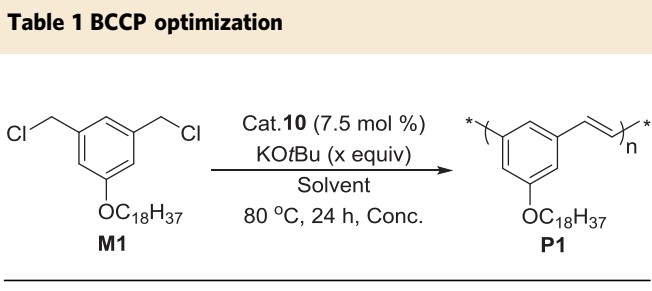

| Entry | Solvent | x= | Conc. (M) | Yield (%) | $M_n$ (Da) | PDI |
|---|---|---|---|---|---|---|
| 1 | CPME | 3 | 0.05 | 91 | 10,400 | 1.28 |
| 2 | MTBE | 3 | 0.05 | 69 | 13,600 | 1.21 |
| 3 | CPME | 3 | 0.1 | 86 | 11,200 | 1.26 |
| 4 | CPME | 3 | 0.2 | 73 | 12,200 | 1.29 |
| 5 | MTBE | 3 | 0.1 | 72 | 15,300 | 1.41 |
| 6 | MTBE | 3 | 0.2 | 71 | 17,400 | 1.42 |
| 7 | CPME | 4 | 0.05 | 92 | 10,600 | 1.23 |
| 8 | CPME | 5 | 0.05 | 92 | 10,300 | 1.22 |
| 9 | CPME | 6 | 0.05 | 91 | 10,200 | 1.22 |

Reactions were conducted using monomer **M1** (0.1 mmol). The products were obtained by reprecipitation from CHCl₃-CH₃OH. Polymer analysis ($M_n$, PDI) were estimated by GPC calibrated on polystyrene standards with THF as eluent

**Fig. 3** Scale-up of **P1**. Synthesis of **P1** by BCCP reaction

$M_n$ 10,600, PDI 1.20
334.8 mg, 90% yield

similar PDIs. The yields, however, dropped from 91% (0.05 M) to 86% (0.1 M) and 73% (0.2 M). With MTBE as solvent, increasing concentration led to higher $M_n$ of 15,300 (0.1 M, entry 5) in 72% yield and 17,400 (0.2 M, entry 6) in 71% yield, with PDI of the corresponding polymers of 1.41. Employing 4 and 5 equivalents of KO*t*Bu afforded polymer product with similar $M_n$, PDI, and yield (entries 7–8). The $M_n$ dropped to 10,200 when 6 equiv. of base were employed (entry 9).

**Scalability of BCCP with monomer** M1. Scalability is an important attribute of polymerization catalysts. We next scaled the BCCP of **M1** to 1 mmol scale using the conditions outlined in Table 1, entry 1. Under the reaction conditions shown in Fig. 3, the polymer **P1** was obtained with $M_n$ 10,600 and PDI 1.20 in 90% yield (334.8 mg).

**Mechanistic distinctness of BCCP**. Traditionally, PPVs were synthesized by a two-step quinodimethane polymerization/elimination protocol (the precursor route)[19]. The quinodimethane polymerization forms a non-conjugated polymer that is then converted to the conjugated PPV by high-temperature (180–300 °C) thermal elimination reaction (Fig. 4a)[49]. The harsh conditions required for converting non-conjugated precursor polymers to PPVs and the incomplete elimination lead to structural defects, which affect the luminescence quantum efficiency of the PPV films[50]. In sharp contrast, sulfenate anion-catalyzed BCCP proceeds by a different mechanism, which results in chemoselective construction of *trans* double bonds (Fig. 4b). Compared with other common methods for the preparation of PPVs such as transition metal-mediated Heck reactions, cross-coupling reactions, and metathesis reactions[51], BCCP is a transition metal-free process.

To highlight the advantage of BCCP over precursor routes (Gilch, Wesling, and Vanderzande methods) (Fig. 4a), we designed monomer **M2** in which two benzyl chloromethyl groups are linked by a C(CF₃)₂ bridge (Fig. 5). As the C(CF₃)₂ linker prevents the formation of quinodimethane intermediate, polymer **P2** could not be prepared by precursor routes (Fig. 4a). Using the conditions in Table 1 (entry 1), the BCCP afforded polymer **P2** in 82% yield with $M_n$ 14,300 and PDI 1.45.

**Synthesis and characterization of alternating 1,3- and 1,4-linked copolymers**. It is well-known that OLED device performance is greatly influenced by the structural regularity of the polymers. To further demonstrate the synthetic potential of BCCP, we next employed BCCP in the synthesis of challenging alternating copolymers. We designed a class of monomers (**M3**, Fig. 6a) by incorporating a flourenyl group between two *meta*-phenyl groups. Polymerization of monomer **M3** bearing different central Ar groups is expected to lead to a new class of structurally regular and alternating poly[(1,3-phenylene vinylene)-alt-arene]s. As proof-of-concept, we synthesized a fluorine-containing monomer **M3-1**. BCCP of **M3-1** led to co-polymer **P3-1** in 91% yield with $M_n$ 13,000 and PDI 2.00 (see Supplementary Method,

against polystyrene standards were performed. In this screen we observed complete polymerization at 10, 7.5, and 5.0 mol % catalyst loadings. There was little impact of the substituents on the aryl ring of the sulfenate anion (ArSO⁻), with similar $M_n$ and polydispersity index (PDI) ($M_n$ ~ 10,000 were observed at 10 and 7.5 mol % loading and $M_n$ ~ 9000 at 5.0 mol % loading). Lower catalyst loadings of 2.5 mol % led to oligomerization (see Supplementary Table 1). Moreover, we observed complete consumption of monomer **M1** after 10 min. A rapid loss of monomer at the beginning of the polymerization indicates that the BCCP proceeds by a step-growth mechanism.

At this stage of our investigations we chose to employ precatalyst **10**. Under the basic conditions of the polymerization, **10** rapidly undergoes E2 elimination to form styrene and generate the sulfenate anion[12]. The most promising results with precatalyst **10** were with 7.5 mol % ($M_n$ 10,900, PDI 1.26). At this loading, we conducted a second screen focused on 10 bases [LiO*t*Bu, NaO*t*Bu, LiN(SiMe₃)₂, NaN(SiMe₃)₂, KN(SiMe₃)₂, NaH, KH, KOSiMe₃, KOPh, NaOMe] under otherwise identical conditions (Fig. 2b). Analysis of the resulting reactions indicated that polymer was obtained only with LiN(SiMe₃)₂, NaN(SiMe₃)₂, KN(SiMe₃)₂ with $M_n$ all lower than with KO*t*Bu from the first screen (see Supplementary Table 2).

The next step in the optimization was a solvent screen. We examined five solvents (THF, dioxane, MTBE (methyl *tert*-butyl ether), toluene, and dimethylformamide). As shown in Fig. 2C, the most promising result was obtained in MTBE ($M_n$ 13,200, PDI 1.23).

After narrowing our optimization parameters to precatalyst **10** (7.5 mol %), KO*t*Bu, and CPME and MTBE as two top solvents, we conducted lab-scale (0.1 mmol) polymerizations to validate the microscale results and further optimize the BCCP (Table 1). Lab-scale polycondensation of monomer **M1** with both CPME and MTBE at 0.05 M concentration yielded polymer with $M_n$ 10,400, PDI 1.28 in 91% isolated yield for CPME and polymer with $M_n$ 13,600, PDI 1.21 in 69% isolated yield with MTBE. The results confirmed that $M_n$ and PDI of polymers obtained at 10 μmol scale could be reproduced at 0.1 mmol scale. With CPME as solvent, increasing concentration to 0.1 M and 0.2 M (entries 3–4) led to higher $M_n$ (11,200 at 0.1 M and 12,200 at 0.2 M) with

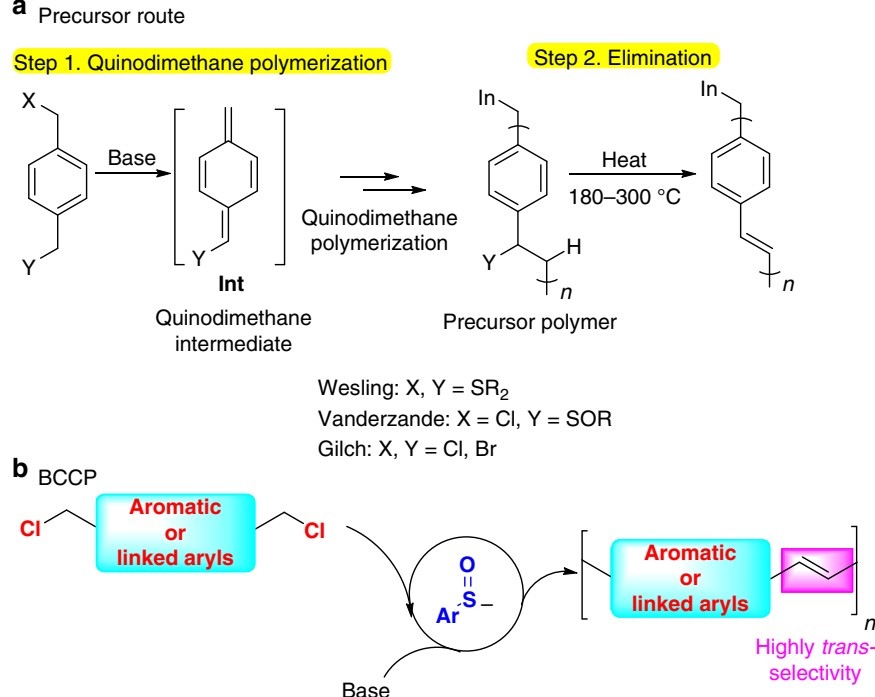

**Fig. 4** Comparison of BCCP polymerization with precursor routes. **a** Summary of precursor route. **b** BCCP polymerization

**Fig. 5** Synthesis of polymer **P2**. The polymer generated has a C(CF$_3$)$_2$ linker

1 mmol Scale synthesis and characterization of co-polymer **P3-1**, **P4-1**, for the synthesis of monomer **M3-1** and co-polymer **P4-1**).

Previous studies have shown that (P$^m$PV$^p$PV)s are highly photoluminescent polymers with well-defined conjugation lengths[52–55]. Such observations inspired us to design a member of this class, monomer **M4**. We expect that polymerization of **M4** type monomers will achieve the synthesis of structurally regular (P$^m$PV$^p$PV)s with 1:1 alternating $^m$PV and $^p$PV units along the chain linked by trans C=C bonds (Fig. 6b). Moreover, incorporation of different moieties as side-chains into the P$^m$PV$^p$PV backbone (R group on the $^p$PV unit of **P4**), would allow tuning of the emission wavelength, emission color and change the quantum efficiency of the resulting co-polymer **P4**. As a proof-of-concept, we synthesized a thiophene-containing monomer **M4-1**. BCCP of **M4-1** led to co-polymer **P4-1** in 93% yield with $M_n$ 8000 and PDI 1.82 (see Supplementary Method, 1 mmol Scale synthesis and characterization of co-polymer **P3-1**, **P4-1**, for the synthesis of monomer **M4-1** and co-polymer **P4-1**). Notably, both polymers **P3-1** and **P4-1** are not accessible by the precursor route, demonstrating the value of BCCP in co-polymer synthesis.

The thermal, photophysical, electrochemical and charge transport properties of **P3-1** and **P4-1** were characterized (see Supplementary Figure 37). As shown in Fig. 7a, I, the decomposition temperatures ($T_d$, corresponding to 5% weight loss) measured from thermogravimetric analysis were 420 °C and 404 °C for **P3-1** and **P4-1**, respectively, indicating good thermal stability. Thermal stability is valuable for long device operation in emission materials[15]. From the glass transition temperatures ($T_g$) observed from differential scanning calorimetry (DSC) for the more rigid fluorene containing **P3-1** was 141 °C; however, no obvious phase transition temperature could be obtained for **P4-1** with more flexible C–C double bonds in the conjugated backbone (Fig. 7b). Both polymers showed bright emission under UV excitation, as shown in the photo images of Fig. 7c,d, II. From the UV-Vis spectra in both solution and film (Fig. 7c,d), it is found that two-dimensional conjugated **P4-1** with the pendant thiophene ring exhibited broader and red-shifted absorption relative to **P3-1**, with a one-dimensional conjugated polymer backbone. Regarding the photoluminescence (PL), the thiophene-containing **P4-1** displayed red-shifted PL compared with **P3-1**, with emission peaks in the deep blue (435 nm in THF and 457 nm in the film state) for **P4-1** and ultra-violet (373 nm in THF and 402 nm in the film state) for **P3-1**. The electrochemical properties were measured by cyclic voltammetry (Fig. 7e). Both polymers exhibited quasi-reversible oxidation and irreversible reduction behavior. The highest occupied molecular orbital (HOMO) and lowest unoccupied molecular orbital (LUMO) energy levels were

**Fig. 6** Design and synthesis of alternating co-polymers. **a** Poly[(1,3-phenylene vinylene)-alt-arene]s. **b** Poly[(1,3-phenylene vinylene)-alt-(1,4-phenylene vinylene)] (P$^m$PV$^p$PV)s

determined from the onset of oxidation and reduction curves for **P3-1** and **P4-1**, and were calculated to be −5.97/−2.60 eV and −5.65/−2.77 eV, respectively. Compared with **P3-1**, the introduction of a strong electron-donating thiophene ring in **P4-1** significantly raises the HOMO level by 0.32 eV, suggesting a more efficient hole-injection and better hole transport properties of **P4-1** in optoelectronic devices. This prediction is in good agreement with the measured hole mobility from the space charge limited current method (Fig. 7f). The hole mobility for the thiophene-containing **P4-1** is estimated to be $1.56 \times 10^{-6}\,\mathrm{cm^2\,V^{-1}\,s^{-1}}$, which doubles **P3-1** of $7.79 \times 10^{-7}\,\mathrm{cm^2\,V^{-1}\,s^{-1}}$.

## Discussion

Introduced herein is a class of organocatalytic polymerization processes termed BCCP. The organocatalysts for this process, sulfenate anions, are operationally trivial to generate from bench-stable sulfoxide precatalysts in the presence of base. Sulfenate anion organocatalysts are unique in that they enable generation of C=C double bonds of the type found in PPV's and other stilbene-based polymers. We demonstrated the application of sulfenate anion-catalyzed transfer polycondensation methods to polymers bearing isolated stilbene motifs. The important conceptual advance of this work is that it suggests that small organic molecules that can activate substrates via nucleophilic attack, acidify neighboring hydrogens leading to umpolung reactivity, and then behave as leaving groups can be considered in polymerization processes to forge C=C linkages. From the synthetic aspect, BCCP offers transition metal-free access to wide varieties of polymers that cannot be synthesized by traditional precursor routes (Gilch, Wesling, and Vanderzande methods). To further demonstrate the synthetic potential of BCCP, two alternating co-polymers were synthesized as representatives of classes of poly[(1,3-phenylene vinylene)-alt-arene]s and (P$^m$PV$^p$PV)s. NMR spectra, thermal, photophysical, electrochemical, and charge transport properties of the above mentioned polymers were characterized. Further studies are underway to apply BCCP to the preparation of novel functionalized polymers.

## Methods

**General procedure for the 0.1 mmol BCCP.** An oven-dried 8 mL microwave vial equipped with a stir bar was charged with monomer **M1** (44.4 mg, 0.10 mmol) under a nitrogen atmosphere in a glove box. A solution of precatalyst **10** (1.73 mg, 0.0075 mmol) in 1.0 mL anhydrous CPME was added by syringe. Next, a solution of KO$t$Bu (33.6 mg, 0.30 mmol) in 1.0 mL anhydrous CPME was added by syringe. The reaction was stirred for 24 h at 80 °C, quenched with 2 drops of H$_2$O via syringe, cooled to room temperature, and opened to air. After the volatile materials were removed with a rotary evaporator, CHCl$_3$ (2 mL) was added into each vial and the slurry solution was allowed to stir for 10 min. Cold methanol (6 mL) and H$_2$O (0.5 mL) was then added into each vial to precipitate the polymer and the slurry solution with polymer suspension was allowed to stir for 10 min. The mixture was then transferred with a pipette onto a Whatman autovial syringeless filter (5 mL, 0.45 μm polytetrafluoroethylene (PTFE) membrane). After the MeOH/CHCl$_3$/H$_2$O solution was filtered, polymer that remained in the filter was washed sequentially with 5 mL MeOH and 5 mL pentane. Finally, the polymer remaining in the filter was transferred into a 20 mL vial with spatula and dried under vacuum to yield a pale yellow solid in 33.8 mg, 91% yield.

**General procedure for the scale-up (1 mmol) polymerization.** An oven-dried 100 mL Schlenk tube equipped with a stir bar was charged with monomer 1 (444.0 mg, 1.0 mmol) and precatalyst **10** (17.3 mg, 0.075 mmol). The Schlenk tube was sealed with a rubber septum and was connected to a Schlenk line, evacuated, and refilled with nitrogen (repeated three times). Next, a solution of KO$t$Bu (336 mg,

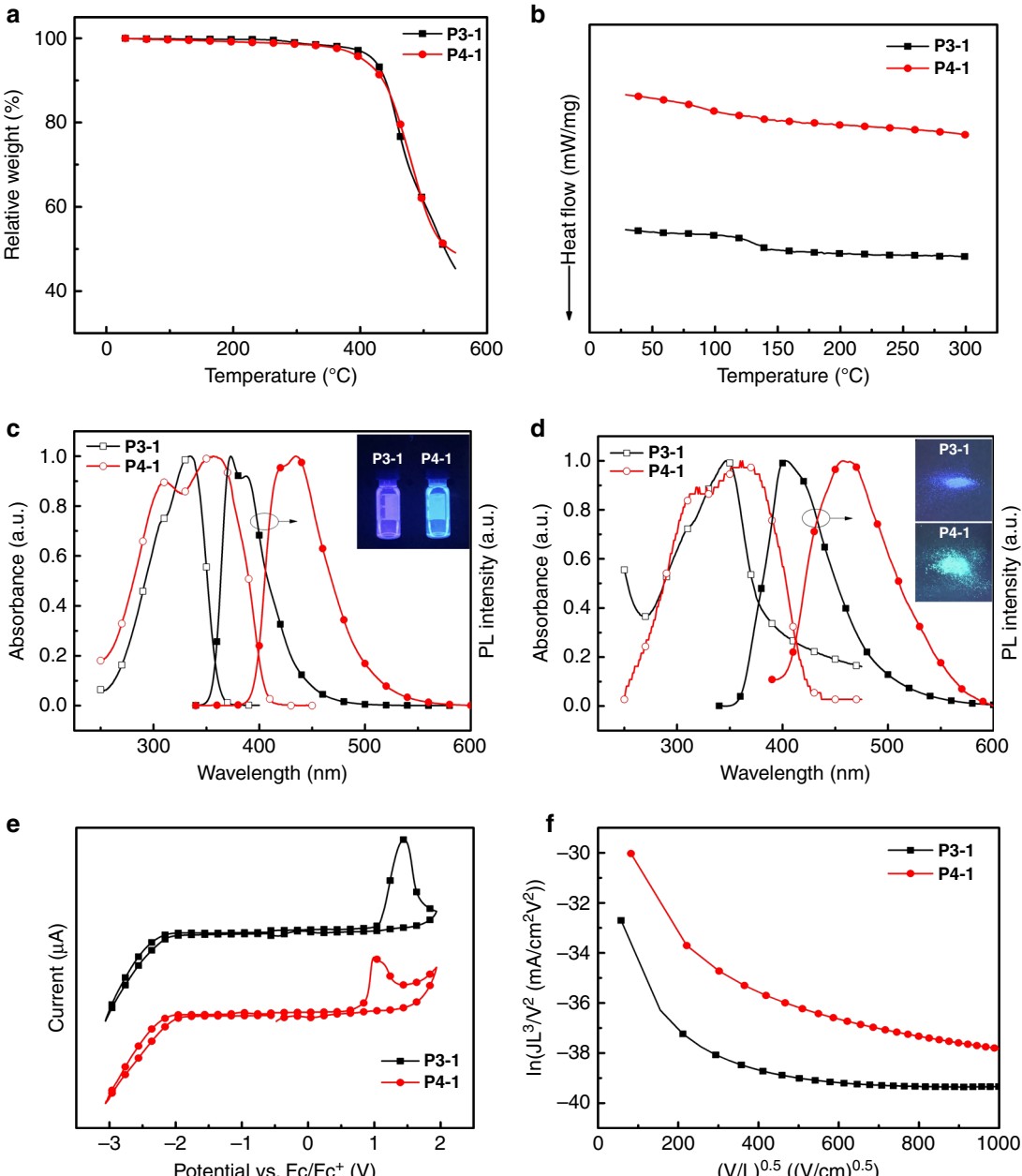

**Fig. 7** Polymer characterization. **a** TGA curves of the co-polymers **P3-1** and **P4-1**. **b** DSC curves of the co-polymers **P3-1** and **P4-1**. **c** Normalized UV-Vis absorption and PL spectra of co-polymers **P3-1** and **P4-1** in THF solution and photographs of polymer photoluminescence under 365 nm light in THF solution. **d** Normalized UV-Vis absorption and PL spectra of co-polymers **P3-1** and **P4-1** in film state and photographs of polymer photoluminescence under 365 nm light in solid state. **e** Cyclic voltammograms of the oxidation curves in dichloromethane and reduction curves in THF for co-polymers **P3-1** and **P4-1**. **f** Current density–voltage (J–V) characteristics of hole-only devices with structure of ITO/PEDOT:PSS(30 nm)/**P3-1** or **P4-1** (30 nm)/MoO₃ (8 nm)/Ag

3.0 mmol) in 20 mL anhydrous CPME was added by syringe. The reaction was stirred for 24 h at 80 °C, cooled to room temperature, opened to air, and quenched with 1 mL of $H_2O$. The reaction mixture was firstly transferred to a 250 mL round-bottom flask and the volatile materials were removed with a rotary evaporator. Next, $CHCl_3$ (20 mL) was added into flask and the slurry solution was allowed to stir for 10 min. Cold methanol (60 mL) was added into the flask to precipitate the polymer and the slurry solution with polymer suspension was allowed to stir for 10 min. The mixture was then filtered on a glass fritted filter funnel (75 mL). After the MeOH/$CHCl_3$ solution was filtered, the resulting solid was washed with $H_2O$ (5 mL), MeOH (20 mL *3), and pentane (5 mL). The solid was collected and dried in a vacuum as pale yellow solid to provide 334.8 mg, 90% yield of the polymer.

**Data availability**. The authors declare that the data supporting the findings of this study are available within the article and its Supplementary Information files.

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

## Acknowledgements

P.J.W. thanks the National Science Foundation (CHE-1464744). We thank Professor Jeung Gon Kim of Chonbuk National University and Dr. Jerome Robinson of UPenn for

helpful discussions. C.W. thanks the Vagelos Integrated Program in Energy Research for funding and support.

## Author contributions

P.J.W. conceived of the project. M.L and P.J.W. designed the experiments and mono-mers. M.L., S.B., C.W., X.Y., Y.L., S.-C.S., B.W., X.F., and R.W. performed the research. Z. L. conducted GPC analysis of co-polymers **P3-1** and **P4-1**. X.G. and Y.T. performed the thermal, photophysical, electrochemical, and charge transport properties characteriza-tion. M.L. and P.J.W. wrote the paper.

## Additional information

**Competing interests:** The authors declare no competing interests.

