## [Peer Review File · Nature Communications]

REVIEWERS' COMMENTS:

Reviewer #1 (Remarks to the Author):

I appreciate the authors efforts to convince me that their method has the impact needed for a more general journal. However, the new polymers given do not in my opinion represent an advance, technically or conceptually, that would justify broadened interest in the community. I think this is a fine piece of catalytic chemistry for the formation of polymers. The polymers produced do not have any particular significance and hence it is really best suited for a more specialized polymer journal.

Reviewer #2 (Remarks to the Author):

In this manuscript, Walsh and coworkers report organocatalytic benzylic-chloromethyl-coupling polymerization (BCCP), which enables the synthesis of meta-linked poly(phenylene vinylenes). Compared to classic precursor routes to PPVs, BCCP is direct, catalytic, and compatible with non-conjugated and meta-linked monomers. The authors have increased the potential impact and broad appeal of this manuscript by synthesizing additional alternating polymers and characterizing their photophysical and electrochemical properties. Given these additions, the quality of this manuscript is comparable to other catalysis articles in Nature Communications. In order for the manuscript to be suitable for publication, the following minor revisions are recommended:

- 1) It is still unclear why the authors have chosen to focus on the less useful meta PPVs rather than the more commonly used para analogues. Other researchers may be interested in using BCCP to make para PPVs. The authors should directly address whether BCCP doesn't actually work in the para case, or if they were specifically interested in meta PPV structures and could only access them by this new chemistry.
- 2) The molecular weights remain limited to less than 20 kDa, even considering the likely overestimation provided by GPC with polystyrene standards. While improving molecular weights may be beyond the scope of this work, if the authors intend to use these materials in devices, it is important to address this limitation. Are the molecular weights limited by solubility, or by termination processes?
- 3) The CV's in Fig. 4 should include the reference electrode.
- 4) ^1H and ^{19}F NMRs of P2 should be provided in the SI.
- 5) The ^1H NMRs of P3 and P4 contain many unexpected small, sharp peaks between 3.25 and 4.75 ppm. Are these part of the polymer or an impurity? If an impurity, what is it? Based on the SI, it seems like a standard polymer precipitation procedure (dropwise addition of a concentrated solution of the polymer in a good solvent to a large amount of the poor solvent) was not used, but rather the inverse order of addition was used. Perhaps a second precipitation or Soxhlet extraction would remove impurities.

Reviewer #2 also looked over reviewer #3 comments and the authors' response.

Point 1: I basically had the same point and they essentially had the same response. I think the current version does address this, although not entirely satisfyingly (see my review); they do have measurements of properties e.g. solid-state emission.

Point 2: I think they took out anything saying that it's large scale, so this is addressed.

Point 3: The new polymers address this comment.

Reviewers' comments:

Reviewer 1:

I appreciate the authors efforts to convince me that their method has the impact needed for a more general journal. However, the new polymers given do not in my opinion represent an advance, technically or conceptually, that would justify broadened interest in the community. I think this is a fine piece of catalytic chemistry for the formation of polymers. The polymers produced do not have any particular significance and hence it is really best suited for a more specialized polymer journal.

Response: We understand that the reviewer does not find the polymers that we have chosen to prepare of great interest. We contend that the introduction of a new method for the synthesis of polymers is of greater importance than the polymers themselves. This method is applicable for the preparation of new families of polymers previously difficult to access.

Reviewer 2:

In this manuscript, Walsh and coworkers report organocatalytic benzylic-chloromethyl-coupling polymerization (BCCP), which enables the synthesis of meta-linked poly(phenylene vinylenes). Compared to classic precursor routes to PPVs, BCCP is direct, catalytic, and compatible with non-conjugated and meta-linked monomers. The authors have increased the potential impact and broad appeal of this manuscript by synthesizing additional alternating polymers and characterizing their photophysical and electrochemical properties. Given these additions, the quality of this manuscript is comparable to other catalysis articles in Nature Communications. In order for the manuscript to be suitable for publication, the following minor revisions are recommended:

1) It is still unclear why the authors have chosen to focus on the less useful meta PPVs rather than the more commonly used para analogues. Other researchers may be interested in using BCCP to make para PPVs. The authors should directly address whether BCCP doesn't actually work in the para case, or if they were specifically interested in meta PPV structures and could only access them by this new chemistry.

Response: The reason we focused on meta PPV's is previously they could not be prepared by Gilch polymerizations or other "precursor routes". They are also difficult to synthesize using other methods, such as Wittig condensation. Our BCCP method offers a direct, organocatalytic approach to such classes of polymers. In short, the choice of meta substituted precursors clearly highlights the utility of catalytic BCCP.

Regarding para-substituted monomers, Gilch mechanism might compete with BCCP with simple 1,4-dibenzyl chloride monomers. However, we are currently synthesizing monomers bearing multi-ring systems and para-chloromethyl groups as illustrated below. Such multi-ring systems will not form quinodimethane intermediates, which are the key intermediates in Gilch-type polymerizations. Preliminary result indicate that BCCP of monomer M5 afforded product P5 in 86% yield with M_n 11800 and PDI 2.65.

We are working to design more monomers and polymers and hope to publish this work after the summer.

2) The molecular weights remain limited to less than 20 kDa, even considering the likely overestimation provided by GPC with polystyrene standards. While improving molecular weights may be beyond the scope of this work, if the authors intend to use these materials in devices, it is important to address this limitation. Are the molecular weights limited by solubility, or by termination processes?

Response: We hypothesize that chain termination process might occur if the benzyl chloride was attacked by KO^tBu via a S_N2 type reaction. This would cause termination of the polymer. We are currently working to address this issue with the goal of publishing a follow up paper in a more specialized journal.

3) The CV's in Fig. 4 should include the reference electrode.

Response: We have updated Fig. 4 with reference electrode.

4) ¹H and ¹⁹F NMRs of P2 should be provided in the SI.

Response: we have added the ¹H and ¹⁹F NMRs of P2 and GPC distribution plots of P2 in the Supplementary Informaiton.

5) The ¹H NMRs of P3 and P4 contain many unexpected small, sharp peaks between 3.25 and 4.75 ppm. Are these part of the polymer or an impurity? If an impurity, what is it? Based on the SI, it seems like a standard polymer precipitation procedure (dropwise addition of a concentrated solution of the polymer in a good solvent to a large amount of the poor solvent) was not used, but rather the inverse order of addition was used. Perhaps a second precipitation or Soxhlet extraction would remove impurities.

Response: Based on the reviewer's comment, we re-did the precipitation by dissolved polymer P3 and P4 in minimum amount of THF followed by dropwise addition to cold methanol. The polymers were collected and NMR is basically the same. Our hypothesis is that these peaks are from polymer chain ends.

For example, in polymer **P3-1**, We propose that peaks around 4.6 ppm are representative peaks for CH₂ of chloromethyl chain end.

Peaks around 4.1 ppm are representative peaks for CH₂ of sulfoxide chain end.

Peaks around 3.8 ppm are representative peaks for methoxyl of sulfoxide chain end.

Reviewer #2 also looked over reviewer #3 comments and the authors' response.

Point 1: I basically had the same point and they essentially had the same response. I think the current version does address this, although not entirely satisfyingly (see my review); they do have measurements of properties e.g. solid-state emission.

Point 2: I think they took out anything saying that it's large scale, so this is addressed.

Point 3: The new polymers address this comment.

We are very grateful to the reviewers for their insightful and helpful suggestions. We hope that the revised manuscript is now suitable for publication in *Nature Communications*.